# Characterization of Host-Associated Microbiota and Isolation of Antagonistic Bacteria from Greater Amberjack (*Seriola dumerili*, Risso, 1810) Larvae

**DOI:** 10.3390/microorganisms11081889

**Published:** 2023-07-26

**Authors:** Vasiliki Paralika, Fotini Kokou, Stelios Karapanagiotis, Pavlos Makridis

**Affiliations:** 1Department of Biology, University of Patras, 26504 Rio, Greece; up1071435@upatras.gr; 2Aquaculture and Fisheries Group, Department of Animal Sciences, Wageningen University, 6700 AH Wageningen, The Netherlands; fotinikokou@wur.nl; 3Galaxidi Marine Farm S.A., 33200 Galaxidi, Greece; s.karapanagiotis@galaxidimarine.farm

**Keywords:** marine fish larvae, antagonistic activity, symbionts, microbial communities, probiotics

## Abstract

Greater amberjack (*Seriola dumerili*) is a new species in marine aquaculture with high mortalities at the larval stages. The microbiota of amberjack larvae was analyzed using 16S rDNA sequencing in two groups, one added copepod nauplii (*Acartia tonsa*) in the diet, and one without copepods (control). In addition, antagonistic bacteria were isolated from amberjack larvae and live food cultures. Proteobacteria was the most abundant phylum followed by Bacteroidota in amberjack larvae. The composition and diversity of the microbiota were influenced by age, but not by diet. Microbial community richness and diversity significantly increased over time. Rhodobacteraceae was the most dominant family followed by Vibrionaceae, which showed the highest relative abundance in larvae from the control group 31 days after hatching. *Alcaligenes* and *Thalassobius* genera exhibited a significantly higher relative abundance in the copepod group. Sixty-two antagonistic bacterial strains were isolated and screened for their ability to inhibit four fish pathogens (*Aeromonas veronii*, *Vibrio harveyi*, *V. anguillarum*, *V. alginolyticus*) using a double-layer test. *Phaeobacter gallaeciensis*, *Phaeobacter* sp., *Ruegeria* sp., and *Rhodobacter* sp. isolated from larvae and *Artemia* sp. inhibited the fish pathogens. These antagonistic bacteria could be used as host-derived probiotics to improve the growth and survival of the greater amberjack larvae.

## 1. Introduction

During the early life stages of marine fish larvae, high mortalities are often observed. Larval mortalities could be caused by disturbances of the microbial balance in the fish, influenced by the microbiomes associated with the water, the live feed, and the rearing system, as well as the interactions between them [1,2]. Consequently, it is important to understand how the microbiome composition is shaped during the early larval stages and its impact on fish survival and growth [3,4]. The diversity of larval microbiomes has been studied using next-generation sequencing (NGS)-based analysis of commercially valuable fish species, such as gilthead seabream (*Sparus aurata*) [5,6], rabbitfish (*Siganus guttatus*) [7], yellowtail kingfish (*Seriola lalandi*) [8], rainbow trout (*Oncorhynchus mykiss*) [9], tilapia (*Oreochromis niloticus*) [10], Atlantic cod (*Gadus morhua*) [11], channel catfish (*Ictalurus punctatus*) [12], and Atlantic salmon (*Salmo salar*) [13]. However, there is still a lack of information about new species in aquaculture, such as the greater amberjack (*Seriola dumerili*).

After hatching, chorion-associated bacteria may become the first colonizers of the developing gastrointestinal tract. The first bacterial colonization of the fish larvae gut primarily comes from the rearing environment [14,15,16], and secondarily, as fish grow, from the diet [17] while other factors, such as ontogenetic development [9], the trophic level [18], and species-specific characteristics [19], also play an important role. After the initial colonization of the larvae gut microbiota, diet influences the gut microbial community and substantial diversification occurs from the first feeding [20,21]. The diversity of bacteria increases as fish develop and the complexity of the gut increases. While the microbial community changes with life stage and habitat, a relatively stable gut microbiota has been reported to be established within the first weeks after hatching [22]. In zebrafish, *Danio rerio*, a core microbial community, is formed via host-selective processes regardless of the environmental parameters [19], which could be related to the maturation of the adaptive immune system [23]. Therefore, knowledge of the larval microbiota composition and the factors that affect it such as the early life diet could be used for microbial community manipulation strategies to improve larval survival.

Selective enhancement of beneficial bacteria in the intensive rearing of the larvae may be achieved by the addition of probiotic bacteria in water or live feed [1]. The use of indigenous probiotic bacteria, isolated from the culture system or the host, could be a potential novel source for health management [24]. The modes of action of probiotic bacteria include the production of inhibitory compounds against pathogenic bacteria, competition for nutrients, or adhesion sites on the gut or other tissue surfaces, and improvement in the rearing water quality [25,26]. The selection of candidate probiotics from the host or the local environment may improve the colonization potential of the gut [27] or improve the survival and growth of the larvae [28,29]. The use of host-associated microorganisms as probiotics with immunomodulatory functions has been tested in salmonids and Atlantic cod [30]. Host-derived probiotics could be a powerful tool for the rearing of fish in critical periods with high mortalities, such as the larval stages of marine fish.

The greater amberjack (*Seriola dumerili*, Risso 1810) is a new species in Mediterranean aquaculture. However, high mortality during its larval and juvenile stages imposes a bottleneck for its domestication. Rotifers and Artemia are commonly used as live food for the larval rearing of this species in marine hatcheries, although none of them forms part of the natural diet of marine fish larvae. Copepods, on the other hand, are part of the natural diet of marine fish larvae, but their routine production is both expensive and unpredictable [31]. The use of copepods in the rearing of greater amberjack larvae resulted in an improvement in bone development and a decrease in the appearance of malformations [32]. Diet may modulate the gut microbiota in fish, and different feeding protocols for the rearing of greater amberjack larvae and juveniles could influence their vulnerability during the early life stages. The characterization of microbial communities during the development of greater amberjack, which is currently lacking, but also under different feeding protocols, may provide new insights into the development of better rearing protocols. Antagonistic bacteria in general comprise bacteria that can outcompete other populations of bacteria in a specific environment by higher affinity to the substrate or attachment sites, production of inhibitory compounds, disruption of intercellular communication, or strong binding to micronutrients [4,25,28]. The isolation of antagonistic bacteria from the “local” environment, with the ultimate target to use them as probiotics, was a secondary target in this work. Therefore, the aims of this study were 1) to describe the host-associated microbiota in greater amberjack larvae fed with two different feeding protocols, and 2) to isolate culturable antagonistic bacteria against fish pathogens from the larvae or the live feed with potential probiotic properties.

## 2. Materials and Methods

### 2.1. Rearing of Greater Amberjack Larvae

Fertilized eggs of greater amberjack (*Seriola dumerili*) were hatched in egg incubators at the facilities of Galaxidi Marine Farm S.A., Greece. Yolk-sac larvae were stocked in four cylindroconical tanks of 3000 L of volume each at about 56 larvae/L, T = 23–24 °C, and pH = 8. The dissolved oxygen saturation level was 85–95%, and water renewal was initiated three days after hatching (dah) and increased thereafter. Lights were turned on at 3 dah at 800 lux intensity, increased to 1500 lux at 15 dah, and reduced to 1000 lux at 20 dah. Two different feeding protocols were applied. Two tanks of the control group were fed with rotifers (*Brachionus* sp.) from 3 to 27 dah, newly hatched *Artemia* sp. nauplii from 10 to 23 dah, enriched *Artemia* sp. metanauplii from 17 dah until the end of the experiment, and formulated feed from 23 dah onwards. Two tanks in the copepod group were additionally fed with newly hatched copepod nauplii (*Acartia tonsa*) 3–17 dah (Figure 1).

### 2.2. Amberjack Larvae Microbiota Characterization 

At each sampling point, two replicates (n = 2) consisting of pooled samples of three larvae were collected from each tank at 9, 15, 23, and 31 dah (Figure 1). The larvae were first anaesthetized with 3-aminobenzoic acid ethyl ester 0.2 mg/mL (MS-222, Sigma, St. Louis, MO, USA), washed with 50 mL sterile seawater using a mesh with 250 μm pore size, and thereafter homogenized in 5 mL filtered sterile seawater in glass homogenizers. Total DNA was extracted from larvae homogenate using Nucleospin tissue (Macherey-Nagel, Germany) DNA, RNA, and protein purification kit following the manufacturer΄s protocol, and analyzed via amplicon sequencing to characterize the microbiota of the homogenized larvae. 

The hypervariable V4 region of the 16S ribosomal RNA gene (16S rDNA) was amplified using primers 515f (5′-CTAGTGCCAGCMGCCGCGGTAA-3′) and 806r (5′-CTAGGACTACHVGGGTWTCTAAT-3′) [33,34]. Sequencing was performed using an Illumina MiSeq Next Generation system (Illumina), following the company’s protocol (MrDNA, Shallowater, TX, USA). Sequencing data can be found in the NCBI (SRA) database under the study accession code PRJNA875121.

An open-source software package, DADA2, version 1.26 [35], was used to model and correct Illumina-sequenced amplicon errors. Data were demultiplexed into forward and reverse reads according to the barcode sequence into sample identity, and trimming was performed [36]. For the forward reads based on the quality profiles, the first 240 nucleotides were kept, and the rest were trimmed. DADA2 resolves differences at the single-nucleotide level and the end product is an amplicon sequence variant table, recording the number of times each exact sequence variant ESV) was observed in each sample (100% sequence identity). Taxonomy was assigned using the Ribosomal Database Project Classifier [37] against the 16S gene reference Silva database (138 version) [38]. Chloroplasts and Mitochondria DNA were removed from the analysis. Due to the variation in sequence depths between samples, all samples were normalized to the lowest depth by subsampling at 29,254 reads per sample. 

For the alpha-diversity analysis, Shannon H′ diversity and richness (observed taxa) were calculated. Non-parametric tests (Wilcoxon test) and linear mixed-effect models (nlme R package) were used to assess alpha diversity [39]. The adonis implementation of Permanova (non-parametric permutational multivariate analysis of variance) was used for the comparison of the beta diversity between groups [40]. Cluster analysis exploring the similarities between microbial community compositions of different samples was examined using Amplicon Sequence Variants (ASVs) abundance (Bray–Curtis metric). The examination of differentially abundant ASVs between groups (sampling days and feed type) was performed using the DESeq2 tool.

We used the PICRUSt2 tool (Phylogenetic Investigation of Communities by Reconstruction of Unobserved States; PICRUSt) [41] to predict the functional content of the fish microbiome originating from the different groups. The ASVs were aligned to the reference sequences and the Nearest Sequenced Taxon Index (NSTI) score was used to evaluate the availability of reference genomes that are closely related to the most abundant microorganisms in the samples. Sequences with NSTI scores >2 were removed from the dataset (49 out of 1434 ASVs), as predictions would be of low accuracy. The functional profile results were then analyzed using the DESeq2 tool.

### 2.3. Bacteria Isolation from Greater Amberjack Larvae and Live Feed 

Serial ten-fold dilutions of the larvae homogenate, at each sampling point, were plated on 90 mm Marine Agar plates (Difco Laboratories, Detroit, MI, USA). The remaining homogenate was stored at −80 °C for the microbiota characterization, as described above. The Petri dishes were thereafter incubated at room temperature (20–22 °C) for 10 days. Colonies that showed inhibition zones of growth were considered antagonistic and were sub-cultured to pure cultures using the streak plate method. These bacterial isolates were preserved at −80 °C in cryovials with ceramic beads (Microbank^TM^ Freezer kit, PRO-LAB Diagnostics, Richmond Hill, ON, Canada). 

Similarly, samples were taken from live food organisms (*Artemia* sp. and rotifers) for microbiological analysis. The sampled live food cultures were washed with 50 mL filtered autoclaved seawater using a mesh with 50 μm pore size and thereafter homogenized in one mL of autoclaved seawater in a glass homogenizer. The homogenates were plated on Marine Agar and the same procedure was followed for the isolation of antagonistic bacteria as previously described for the fish larvae samples.

### 2.4. Bacterial DNA Extraction and PCR Amplification

Bacterial isolates were later regenerated in test tubes with marine broth and thereafter spread on Marine Agar to evaluate the purity of the cultures. Bacterial cells (1:20) were heated in distilled water at 98 °C for 15 min and centrifuged for 10 min according to Jensen, Bergh, Enger, and Hjeltnes [42]. The universal bacterial primers 27f and 1492r were used to amplify the 16S ribosomal genes [43]. PCR was performed in 50 μL reaction mixtures comprising 0.05U Taq polymerase (Promega, Madison, WI, USA), 2.5 mM MgCl_2_, 1× buffer, 200 μM dNTPs, 1 μL diluted cell suspension, and 0.5 μM of each of the 27f and 1492r primers. Reactions were carried out in 50 μL reaction mixtures with an initial denaturation step of 95 °C for 15 min followed by 30 cycles of 92 °C for 1 min, 55 °C for 1 min, and 72 °C for 45 s; the final extension step was performed at 72 °C for 5 min. PCR products were purified using a DNA purification kit (Qiagen, DNeasy Tissue Kit, Venlo, The Netherlands) were quantified using a NanoDrop spectrophotometer and visualized by 1% agarose/TAE gel electrophoresis. The samples were sent for sequencing at Macrogen Europe, The Netherlands.

### 2.5. Inhibition Tests

The bacteria isolated from the larvae and live feed homogenates were screened in vitro for antagonism against four fish pathogenic bacteria using a double-layer approach [44]. The pathogens were *Vibrio anguillarum* type strain LMG 4437, isolated from Atlantic cod (*Gadus morhua* L.) by Dr. J. Bagge [24]; *Vibrio alginolyticus* type strain V2 isolated from *Dentex dentex*, during outbreaks of vibriosis [45]; *Vibrio harveyi* type strain VH2, isolated from farmed juvenile *Seriola dumerili* during outbreaks of vibriosis in Crete, Greece [46]; and *Aeromonas veronii* biovar sobria isolated from farmed European seabass in the Mediterranean Sea [47]. The isolates were kindly provided by Dr Pantelis Katharios from the Hellenic Center for Marine Research, Heraklion, Crete, Greece. Briefly, the bacteria tested were first cultured in 5 mL of tryptic soy broth (TSB) supplemented with 2% NaCl (TSBS) at 25 °C. After 2 days, a 5 μL drop was transferred to the center of a plate with TSAS, which was thereafter incubated for 2–3 days at 20 °C. When the colonies appeared, the dishes were exposed to chloroform vapor for 20 min. A second layer of semisolid TSAS (8 gr agar/L of tryptic soy broth supplemented with 2% NaCl) was poured on each plate where the pathogenic strain had been previously inoculated. For each test, the inhibition was considered positive if a clear zone area (inhibition halo) was apparent around the colonies, and inhibition zones were measured at the biggest distance (diameter) of the inhibition zone in mm. The double-layer test against the four fish pathogenic strains was repeated using Marine Agar as a culture medium for the antagonistic bacterial strains. 

## 3. Results

### 3.1. Microbiota Characterization of the Greater Amberjack Larvae

The microbial richness, as measured using linear mixed-effect model analysis, significantly increased over time (*p* < 0.05; Table 1), starting from a lower richness during early larval stages at 9 dah, and increasing gradually until 31 dah (Figure 2A). The lowest ASV richness occurred at 9 dah (145 ± 53) and the highest at 23 dah (290 ± 119) for the control and 31 dah (314.5 ± 92.5) for the copepods group, respectively. For the microbial diversity using Shannon H indices, which consider not only the number of bacterial phylotypes (i.e., ASVs) but also their relative abundances in each sample, significant effects were observed by the age of the fish (*p* < 0.05; Table 1). A gradual increase in both richness and diversity was observed during the experiment indicating an increase in microbial colonization in the larvae (Figure 2B). The dietary treatment had no significant effect on either the microbial richness or diversity.

Beta-diversity (microbial composition) of the bacterial communities of amberjack larvae was significantly influenced by age (two-way Permanova analysis; *p* < 0.001; Table 2), while no significant effect of the feeding protocol was shown (*p* > 0.05; Figure 3).

Two bacterial phyla, Proteobacteria and Bacteroidota, accounted for 96.4% of all the retrieved quality filtered sequences. Proteobacteria was the dominant phylum, with an average relative abundance of 85.6% across all the samples followed by Bacteroidota (10.79%), Firmicutes (0.77%), Campilobacterota (0.68%), Actinobacteriota (0.47%), Planctomycetota (0.36%), and Patescibacteria (0.34%) (Appendix A). Amberjack larvae were mostly characterized by bacteria belonging to the Rhodobacteriaceae family with a mean relative abundance for all sampling points at 50.25%, along with the families Vibrionaceae and Flavobacteriaceae, showing respective percentages of 12% and 9.6% (Figure 4), followed by Alteromonadaceae (4.8%), Pseudoalteromonadaceae (2.5%), and Hyphomonadaceae (2%). There was a difference in relative abundances of families between the two groups in 9 dah larvae, as 13 families in larvae from the copepods group, and only 5 families from the control group showed relative abundance greater than 1%. So, a higher diversity was found in the copepods group at the family level. Additionally, the 9 dah larvae from the copepods group appeared to have the highest relative abundances of families from all sampling points, such as Pseudoalteromonas (11.6%), Alcaligenaceae (5.9%), Colwelliaceae (5%), Saccharospirillaceae (3.7%), Oleiphilaceae (3.6%), and Comamonadaceae (2.3%). 

A large proportion (average relative abundance at 15%) of the microbial communities was not assigned to a specific genus, whereas the most abundant genus was *Thalassobius,* exhibiting relative abundance at a mean percentage of 16,5% across all samples, followed by *Vibrio* (12%), *Dokdonia* (8.4%), *Cognathshimia* (7,9%), *Ruegeria* (7.5%), and *Alteromonas* (4.5%) (Appendix A). In 9 dah larvae from the copepods group, the genera as Pseudoalteromonas, Alcaligenes, Thalassotalea, Oleiphilus, and Neptuniibacter showed their highest relative abundances at 9.4%, 5.9%, 5%, 3.67%, and 3.5%, respectively, across the experimental period, following the Shannon diversity index, which was higher for larvae from the copepods group compared with the control group (*p* < 0.05).

The core microbiota is defined as any set of microbial taxa, as well as the associated genomic or functional attributes characteristic of a specific host or environment [48]. To determine the presence of an ASV in a group, its prevalence was set to be higher than 50%, meaning that it was present in at least one sample within each tank and showed 1% of minimum relative abundance. The core microbiota of the two groups was similar (Appendix A). *Marinomonas* (ASV45) and *Thalassococcus* (ASV30) were unique to the control and the copepods group, respectively. Nine ASVs were present in the larvae from both feeding protocols at all sampling points. The shared core ASVs belonged to the genera *Thalassobius*, *Ruegeria*, *Nautella, Dokdonia*, *Donghicola*, *Cognatishimia*, *Vibrio*, and *Alteromonas*; and one ASV was unassigned. In addition, we found significantly enriched taxa in the copepod feeding protocol’s larvae. Specifically, the genus *Alcaligenes* exhibited a significant (*p* < 0.05) differential relative abundance of about 6% in the copepods group at 9 dah (Appendix A). The genus *Thalassobius*, which belongs to the core genera, was the most abundant genus at all sampling points, and its relative abundance was detected significantly (*p* < 0.05) higher for the larvae from the copepod feeding protocol larvae (Appendix A).

Furthermore, to determine the predictive functionality of larvae microbiota for different feeding protocols, functional assessment using PICRUSt2 was performed and assigned to the KEGG (Kyoto Encyclopedia of Genes and Genomes) pathways. One pathway was significantly (*p* < 0.05) enriched in larvae from the copepods group, related to the perosamine synthetase (KEGG Orthology: K13010), which is related to amino sugar and nucleotide sugar metabolism (Figure 5). 

### 3.2. Bacterial Strains Isolated from Greater Amberjack Larvae and Live Feed 

A total of 62 antagonistic culturable bacterial strains were isolated, 59 from larvae homogenates and 3 from homogenated *Artemia* sp. nauplii. The partial length of the 16S rRNA gene was sequenced to identify the isolated bacterial strains. The bacterial isolates were identified at the family level, and 45 bacterial strains were members of Vibrionaceae, the most abundant antagonistic bacteria. The rest were nine Rhodobacteraceae, three Pseudoalteromonadaceae, three Alteromonadaceae, and one annotation was found for each of the Halomonadaceae and Moxarellaceae families (Figure 6).

Among the isolated strains, there were several putative probiotics belonging to the Rhodobacteraceae family, as shown in Table 3. The isolate from the Halomonadaceae family matched with the *Halomonas* sp. strain.

### 3.3. In Vitro Inhibition Tests against Pathogens

The first trial of the double-layer test for the 62 bacterial isolates in vitro inhibition against four fish pathogenic strains, using TSA as the culture medium of the bacterial strains, showed that 12 of them inhibited the growth of the pathogens. Almost all the bacterial isolates from fish belonged to the Vibrionaceae family (Table 4). The twelfth bacterial strain, *Halomonas* sp., which was isolated from homogenized *Artemia* sp., exhibited inhibition zones greater than 30 mm in tests with *V. alginolyticus* and *V. harveyi*.

The double-layer test that was performed for the isolated bacterial strains, using Marine Agar as a culture medium, showed that four strains exhibited antagonistic activity against the fish pathogens (Table 5). These bacterial strains were *Phaeobacter gallaeciensis* strain JL2886, *Phaeobacter* sp. R-52698, and *Ruegeria* sp. InS-296, which was isolated from amberjack larvae, and one bacterial strain, *Rhodobacter* sp. 1–5, which was isolated from homogenized *Artemia* sp. The strain *Phaeobacter gallaeciensis* strain JL2886 was isolated from homogenized greater amberjack larvae of 9 dah, the strain *Phaeobacter* sp. R-52698 was isolated from the larvae of 16 dah (both from the copepods group), and the strain *Ruegeria* sp. InS-296 was isolated from larvae of 16 dah (control group).

## 4. Discussion

Greater amberjack is a new species for European aquaculture with a high economic value [49]. Despite intensive research on the rearing of amberjack larvae, high mortalities are often observed, which is a bottleneck for the commercialization of this species. This is the first report to characterize the larval microbiome of this species using next-generation sequencing. 

The host microbiome may influence the physiology, digestion, and development of the immune system in marine fish larvae [50]. Understanding host-microbiota interactions could be essential for the improvement of rearing protocols or microbial management in larval cultures, for example, using probiotic bacteria. 

### 4.1. Bacterial Dynamics Associated with the Early Developmental Stages of Greater Amberjack Larvae

During the experiment, we observed a gradual change in the alpha diversity of the larval microbiota. Bacterial community richness and diversity (measured as Shannon H’) significantly increased over time (Figure 2A,B). In gilthead seabream, the lowest bacterial richness was found at 1–5 dah [5,6], whereas in cod larvae, the lowest bacterial richness was observed at 17 dah [11]. Regardless of the host species, rearing system, or feed regime, the bacterial diversity in fish larvae tends to increase gradually with age until it reaches a peak when approaching the juvenile stage [2]. Interindividual variation in the gut microbiota composition has been observed in the microbiomes of other fish species, such as zebrafish [51] and Gibel carp [52]. In this study, age was the most significant factor shaping the microbiota in greater amberjack larvae, creating different clusters between 9, 15, and 23 with 31 dah (Figure 3). During the early developmental larvae stages (9 dah and 15 dah), regardless of the diet, there is a stage-specific bacterial community as a result of physiological changes such as gut differentiation, the appearance of acidic conditions in the stomach, or development of the immune system [13,53,54,55]. Generally, the microbiota of fish larvae comprises four dominant phyla: Proteobacteria, Bacteroidota, Firmicutes, and Actinomycetes, with changes in dominance, observed depending on the fish species, developmental stage, rearing system, and feed, as shown in the case of Atlantic cod [11], yellowtail amberjack [8], and gilthead seabream larvae [5]. We observed the same dominant phyla in the greater amberjack larvae. Bacteriodota showed the highest fluctuation in relative abundance among these phyla, with proportions of 4%, 18.6%, 14.4%, and 6% at 9, 15, 23, and 31, respectively. Firmicutes abundance remained low during the whole experiment (0.1–0.6%), reaching their highest average relative abundance (2.23%) in larvae at 31 dah. The presence of Firmicutes has been reported during the weaning process to formulated diet in many marine species [3,56,57] which can explain why the abundance of Firmicutes increased at the last sampling point as commercial feed was provided to the larvae after 23 dah. So, as reported in studies with other fish species [5,8,11], the microbiota of the greater amberjack can be influenced by the type of feed provided to the fish. 

At the family level, Rhodobacteraceae dominated and was the most abundant family during the whole experiment (28.9–72.2% mean relative abundance), followed by Vibrionaceae and Flavobacteriaceae. The family Rhodobacteraceae colonizes tank wall biofilms and comprises predominately Κ-strategist species [58,59]. Members of this family showed probiotic activity by producing antimicrobial substances against Vibrios [27,60,61]. Recently, the isolation of the putative probiotic *Phaeobacter* sp. from marine fish yolk-sac larvae was reported for the first time [24]. In our study, the presence of dominant host-associated microbiota with probiotic characteristics was important, as they may have a greater chance of colonizing the gut and thereby confer a health benefit to the host. 

*Thalassobius* of the Rhodobacteraceae family was the most abundant genus (16.5%), with its relative abundance being at the same level for both groups at the early larval stages but higher in larvae 31 dah from the copepod group. Members of the genus *Thalassobius* have been isolated from marine environments, particularly from surface coastal seawater and tidal flat samples [62]. Independent of the water treatment system used in the hatcheries, the uptake of bacteria by larvae is dominated by bacteria entering with the live food [55]. However, seawater microbiota also represents a significant contribution since marine fish drink seawater to osmoregulate, so bacteria from the water colonize the digestive tract before active feeding commences [16]. The outer surfaces of the fish larvae and the gills are colonized by bacteria in the rearing water. In addition, all live food organisms are filter-feeders, so to filter the water for food particles, they ingest free-living bacteria from the water. These bacteria may accumulate in the live food organisms and be in turn ingested by the fish larvae. Therefore, the water microbiota has a strong influence on the gut during early life stages [63]. The high abundance of *Thalassobius* in the microbiota of amberjack larvae in this experiment could be related to either drinking water by the larvae or the consumption of live feed organisms, since it appears to increase after 9 dah, indicating a strong influence of the live feed microbiota. 

### 4.2. Bacterial Composition and Functionality Associated with Different Feeding Protocols in Greater Amberjack Larvae

In our study, it was shown that the age of the larvae had a strong impact on the microbiota of greater amberjack larvae. In contrast, the feeding protocol had only a minor influence on the microbial communities. The addition of copepods led to only a slight increase in the richness and diversity of the larval bacterial community, mainly at 9 dah and 31 dah. More specifically, the larvae microbiota fed either diet was colonized by Proteobacteria and Bacteroidota, sharing approximately 96% of the relative abundance at each sampling point. Both phyla showed the same pattern of fluctuation between the sampling points for the two diets. Interestingly, the relative abundance of Firmicutes differed at 31 dah, with 3.12% for the copepod and 1.34% for the control group. 

At the family level, Rhodobacteraceae was the dominant family in both groups but showed a different pattern of relative abundance. Specifically, larvae from the control group showed a gradual decrease from 72.2 to 33.2% over time. On the contrary, the larvae from the copepod group showed a gradual increase from 28.9 to 71% over time. Additionally, while in larvae from both groups, the relative abundance of Rhodobacteraceae was higher compared with Vibrionaceae during the whole experiment, at 31 dah larvae from the control group Vibrionaceae showed similar abundance with Rhodobacteraceae. Members of the Vibrionaceae family are widely found in marine environments, often isolated from the intestinal tract of various marine fish species at the larval and fry stages, such as seabass, turbot, Atlantic cod, Atlantic halibut, and Dover sole [50].

Regarding the presence of *Vibrios* in the larvae microbiota, due to the feeding protocols, there was a difference in their average relative abundance being 17% for the control (the most abundant genus in this group) and 7% for the copepod group, respectively. Larvae from the control group collapsed at 31 dah, whereas the larvae of the copepods group survived up to 42 dah (Figure 1). At 31 dah, in larvae from the control group, the relative abundances of Vibrionacae and Rhodobacteriaceae families were 34.3% and 33.3%, respectively. The proportions of the same families for larvae of the same age from the copepods group were 5.5% and 71%, respectively. The feeding regime did not influence the richness and diversity of the larval host-associated microbiota, but it seemed that the addition of copepods resulted in higher survival of the larvae, which could be due to the restriction of Vibrionaceae, and enhanced presence of Rhodocteriaceae, revealing antagonism against Vibrio species and possible probiotic activity. 

Larvae at 9 dah from the copepods group showed the highest abundances of Pseudoalteromonas, Alcaligenaceae, Colwelliaceae, Saccharospirillaceae, Oleiphilaceae, and Comamonadaceae, families, except Vibrionaceae and Rhobocteriaceae, which resulted in higher diversity than the control group at this age. Fish fed a variety of feed items may support more diverse gut-associated bacterial communities because of a wider range of potential substrates available to the bacteria [17]. *Thalassobius* was the significantly dominant genus in the greater amberjack larvae microbiota. Copepods have higher protein and free amino acid content compared to Artemia and rotifers [64]. The protein content of copepods is on average 50% higher than that of the protein content of Artemia [31]. Thus, the need of the larvae fed with copepods to adapt to greater amounts of specific dietary elements, such as amino acids, might lead to stable colonization of the larval microbiota of the genus *Thalassobius,* as this genus is strictly aerobic and uses organic acids and amino acids as carbon sources [65]. The high dietary value of the copepods in the greater amberjack larvae microbiota is supported by the increase in metabolic pathways related to amino acid and glycan biosynthesis in the copepod-fed larvae, indicating a potential nutritional benefit when these live feed organisms are included in the diet of amberjack larvae. Moreover, the genus *Alcaligenes* showed a significantly higher relative abundance in the copepod group at 9 dah, indicating that this genus may play an essential role in specific larval adaptations as an adaptation to the copepod biochemical composition. *Alcaligenes* have been recognized as bacterial species isolated from the intestinal tracts of marine fish species at the larval and fry life stages, specifically in Dover sole and turbot [50]. In agreement with our findings, Alcaligenes have been isolated from turbot larvae fed with copepods [66].

In this study, the experiment was run under industrial conditions. The larvae had to be fed according to the routine operations of the hatchery and could not be starved. The larvae were not disinfected because of the need to isolate antagonistic culturable bacteria and were only rinsed with autoclaved seawater. Due to the small body size of the larvae, dissection of the gut was not possible at the sampling points. Thus, the microbiota of greater amberjack larvae described in this study represents bacteria originating from the skin, gills, and gastrointestinal tract.

### 4.3. Isolation of Cultivable Antagonistic Bacteria with Probiotic Effect

In this study, culturable antagonistic bacteria were isolated from homogenates of greater amberjack larvae and live food, using a novel technique. This method was based on the appearance of inhibition zones in Petri dishes spread with serial dilutions of the homogenates. In this way, it was possible to recognize and isolate bacterial strains with antagonistic activity against other bacteria in the same sample. 

In total, 62 antagonistic bacterial strains were isolated; 59 from larvae homogenates and 3 from homogenated *Artemia* sp. Sequencing and subsequent annotation revealed that the most numerous antagonistic bacterial strains, coming from the larvae, belonged to the Vibrionaceae family (72%). *Vibrio* is a genus of Gram-negative bacteria that is highly salt-tolerant and unable to survive in freshwater; consequently, it is commonly found in various saline or brackish water environments. Among the Vibrionales order, the Vibrionaceae family comprises aquatic bacteria, mostly living in warm waters that tolerate different levels of salinity, including fresh, brackish, and marine waters. The Vibrionaceae family includes the genera *Aliivibrio*, *Catenococcus*, *Enterovibrio*, *Grimontia*, *Listonella*, *Photobacterium*, *Salinivibrio*, and *Vibrio* [67], and it is a genomically, phylogenetically, and functionally diverse group. In agreement with our findings, *Vibrio* strains have been isolated from the intestinal tracks of marine fish species at larval, juvenile, and adult life stages (seabass, turbot, Atlantic cod, Atlantic halibut, and Dover sole) [50]. Vibrionaceae have been reported to be the most numerous antagonistic genera associated with fish, such as Japanese flounder (*Paralichthys olivaceus*) [68], and Senegalese sole [44], but also live feeds (rotifer, Artemia, and copepods) [32]. 

In addition to the members of the Vibrionaceae, representatives from Pseudoalteromonadaceae and Roseobacteraceae families were isolated in proportions of 5% and 14%, respectively, among the antagonistic bacterial strains. *Pseudoalteromonas* isolates have been used as putative probiotics in different farmed organisms. For instance, *Pseudoalteromonas* strains isolated from the intestinal tract of Atlantic cod, the gonads, and the intestinal content of *S. lalandi*, demonstrated antagonistic activity against the pathogen *Vibrio anguillarum* [60], and improved larval growth and survival in *S. lalandi*, [69]. These results are in agreement with our findings that Pseudoalteromonadaceae are a part of the greater amberjack larvae microbiota demonstrating antagonistic activity and are putative probiotics.

Probiotics as an alternative to the use of antibiotics and other chemotherapeutic agents can make aquaculture more sustainable and decrease the ecological footprint of the sector, as they have the potential to improve the GI tract of the fish, including development and maturation of the intestine and immune system [70,71], and resistance to infectious pathogenic microbiota [72]. Three Roseobacteraceae bacterial strains were isolated, which were identified as *Phaeobacter gallaeciensis*, *Phaeobacter* sp., and *Ruegeria* sp. *Phaeobacter gallaeciensis* was isolated from 9 dah and *Phaeobacter* sp. from 15 dah, larvae from the copepod group, while *Ruegeria* sp. was isolated from 9 dah larvae from the control group. These strains showed inhibitory activity against *Vibrio anguillarum*, *Vibrio alginolyticus*, *Vibrio harveyi* and *Aeromonas veronii* strains in vitro. Among others, *Phaeobacter* species have been shown to reduce the pathogenic load in cultures of microalgae and cod larvae during larviculture [73,74] and increase larval survival and specific growth rate in sea bass larvae [24]. Several *Phaeobacter* strains have been isolated in marine aquaculture from the seawater and collectors of scallops (*Pecten maximus*) [75], larval rearing tank walls [67], and different units of Danish turbot farms [76]. As in Makridis et al. [24], *Phaeobacter* strains were isolated from homogenated greater amberjack larvae. *Phaeobacter* has been isolated from tank walls or biofilters of the rearing systems, which are aerobic environments, but from the reared larvae. As they were isolated from greater amberjack larvae at the early developmental stages (9 and 15 dah), it appears that they colonized the surface of the larval body, gills, or gut. The sampled amberjack larvae were not surface-sterilized but only washed with sterile water, so *Phaeobacter* could be isolated from these aerobic surfaces, or they could be a transient part of the gut microbiota.

Three antagonistic bacterial strains were isolated from *Artemia* sp., *Halomonas* sp., *Psychrobacter* sp., and *Rhodobacter* sp. The isolated *Halomonas* sp. showed very strong antagonistic activity against *Vibrio alginolyticus* and *Vibrio harveyi*, while *Rhodobacter* sp. showed smaller inhibition zones against *Aeromonas veronii* and *Vibrio harveyi*. In previous studies, *Psychrobacter* sp., isolated from the whole intestine of juvenile grouper *E. coioides* and as a probiotic when it was administrated with the diet, improved the autochthonous diversity along the gastrointestinal tract of this grouper [77]. It has also been suggested to be capable of producing and secreting effective antimicrobial substances [78]. Following the previous studies, we hypothesize that the isolated strains in our study can be important probiotic candidates for future studies. 

## 5. Conclusions

In our study, we found that greater amberjack larvae reared in a commercial hatchery were colonized by a diverse range of bacteria and that the composition of the host microbiota is mainly influenced by age rather than by diet. Despite this, the slight increase in diversity together with the decrease in potential opportunistic groups such as *Vibrios* could indicate a positive enrichment of the copepods in the greater amberjack larvae microbiota. Finally, we were able to isolate bacteria from the larvae with antagonistic activity against fish pathogens, mainly from the family Rhodobacteraceae, suggesting that the host-associated microbiota is a good source of probiotics.

## Figures and Tables

**Figure 1 microorganisms-11-01889-f001:**
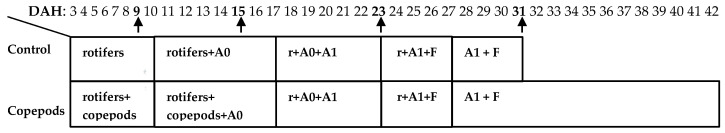
Schematic drawing of experimental feeding protocols of greater amberjack larvae during the experimental period (3–42 DAH) (feeding protocols: control and copepods; r: rotifers; A0: newly hatched *Artemia* sp. nauplii; A1: enriched *Artemia* sp. metanauplii; F: formulated feed). Arrows indicate the sampling days.

**Figure 2 microorganisms-11-01889-f002:**
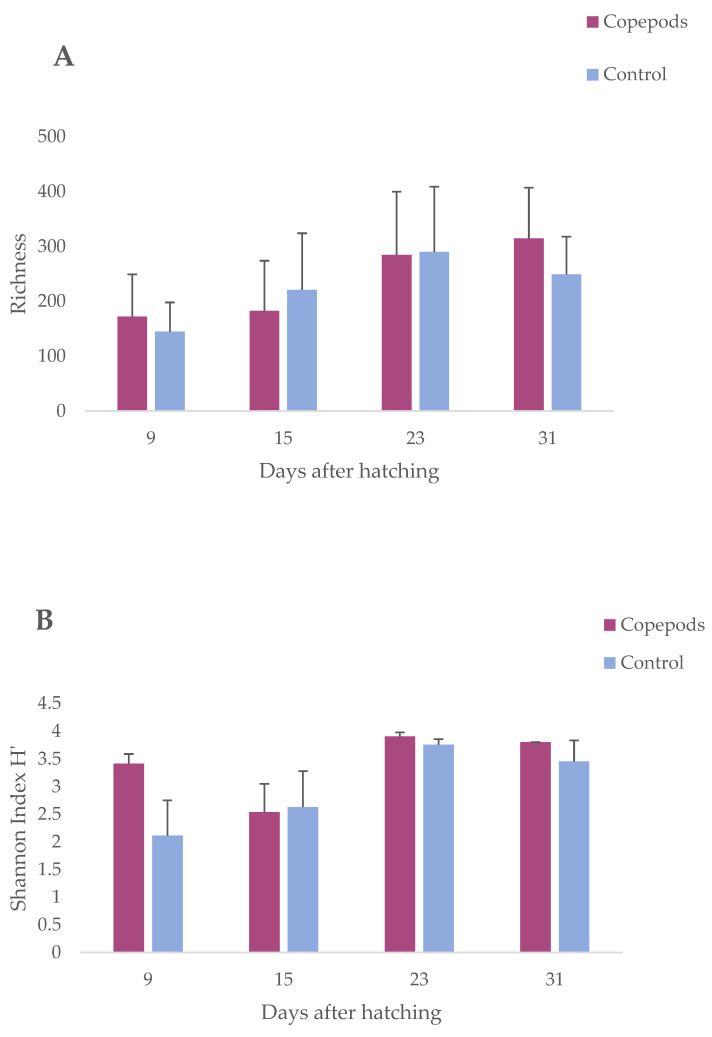
(**A**) Microbial richness during greater amberjack larvae development as indicated by linear mixed-effect model analysis (group means ±SEM); (**B**) Shannon’s diversity indices (±SEM) for microbial communities associated with pooled grater amberjack larvae samples between the control and the copepod groups 9, 15, 23, 31 dah.

**Figure 3 microorganisms-11-01889-f003:**
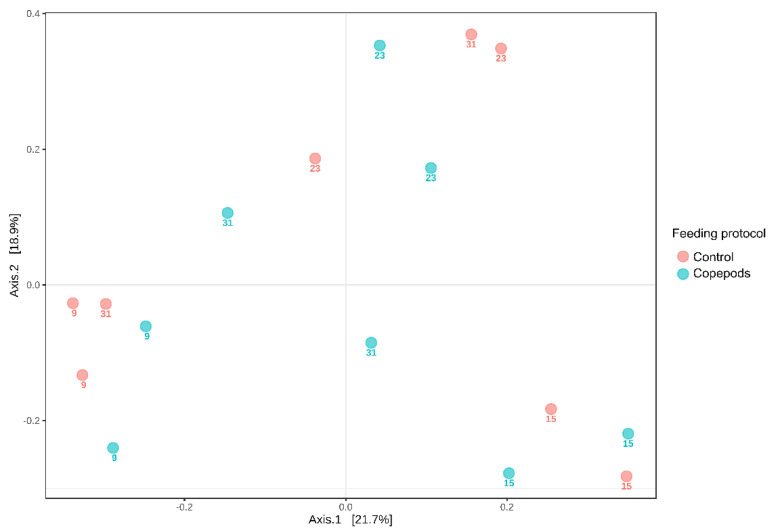
Principal coordinate analysis using Bray-Curtis as dissimilarity metric, indicating clustering (two-way Permanova, *p* < 0.05) due to age (numbers) and not feeding protocol (colours) (days after hatching: 9, 15, 23, 31 dah).

**Figure 4 microorganisms-11-01889-f004:**
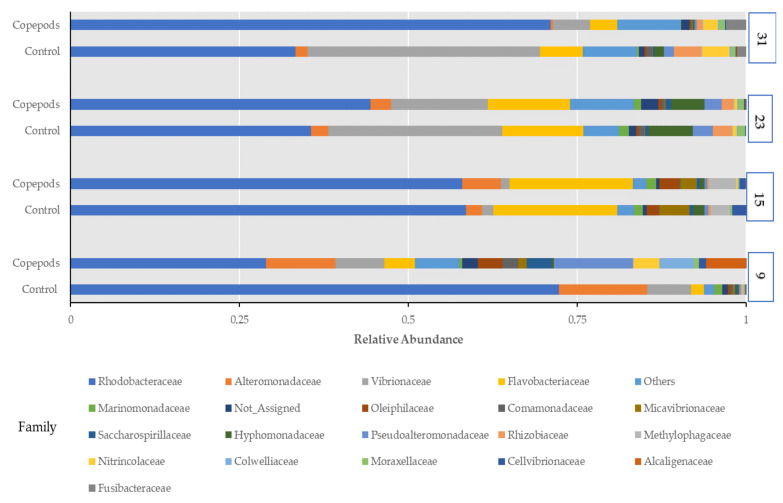
Relative abundance of prokaryotic families in all sampling points (control, copepods; diet) (9, 15, 23, and 31 dah).

**Figure 5 microorganisms-11-01889-f005:**
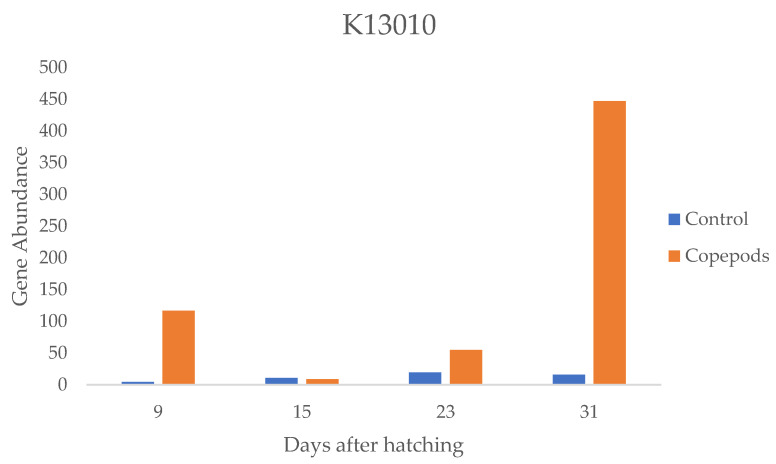
Gene abundance associated with the perosamine synthetase (K13010) pathway in larvae 9, 15, 23, 31 dah in the two groups (control, copepods).

**Figure 6 microorganisms-11-01889-f006:**
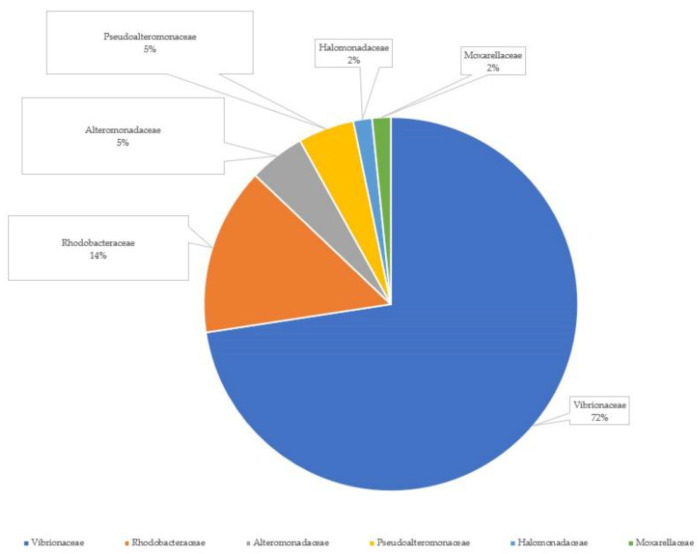
Family-level summary of BLASTN results in proportions for the identified 62 bacterial strains isolated from homogenated greater amberjack larvae coming and homogenated *Artemia* sp.

**Table 1 microorganisms-11-01889-t001:** Linear mixed-effects model using restricted maximum likelihood (RELM) for age and diet effects on richness and diversity.

Richness			
AIC = 165.4049, BIC = 168.3143, logLik = −76.70243			
	den DF	F	*p* value
Age	11	10.39	0.008
Diet	11	0.72	0.41
Age Diet interaction	11	1.51	0.24
Diversity (Shannon Index H’)			
AIC = 50.44739, BIC = 53.35683, logLik = −19.2237			
	den DF	F	*p*-value
Age	11	9.82	0.009
Diet	11	1.27	0.28
Age Diet interaction	11	2.32	0.15

**Table 2 microorganisms-11-01889-t002:** Effect of the diet and time on microbial community profiles by two-way Permanova.

	Sum of Square	Degree of Freedom	Mean Square	F	*p*
Diet	0.18	1	0.18	0.9	0.54
Age	1.80	3	0.60	2.97	0.0001
Interaction	0.53	3	0.18	0.87	0.66
Residual	1.62	8	0.20		
Total	4.14	15			

**Table 3 microorganisms-11-01889-t003:** Isolated bacterial representatives from Rhodobacteraceae family according to the BLASTN results (^1^ isolated strains from homogenated larvae, ^2^ isolated strains from homogenated *Artemia* sp.).

Bacterial Strain	Accession Number	Identity (Pct %)	E Value	Bit Score
*Phaeobacter gallaeciensis* JL2886 ^1^	CP015124.1	95	0	2111
*Phaeobacter* sp. ^1^	KT185144.1	96	0	2159
*Ruegeria* sp. InS-296 ^1^	MF359524.1	96	0	2130
*Ruegeria* sp. InS-119 ^1^	MF359371.1	97	0	2200
*Ruegeria* sp. InS-264 ^1^	MF070517.1	99	0	2350
*Nautella italica* (*Phaeobacter italicus*) ^1^	HQ908722.1	97	0	2196
*Roseobacter* sp. ^1^	KY770280.1	97	0	2233
*Rhodobacteraceae bacterium ZJ3003 * ^1^	KP301108.1	98	0	2281
*Rhodobacter* sp. 1–5 ^2^	AF513400.1	91	0	1661

**Table 4 microorganisms-11-01889-t004:** Antimicrobial activity against pathogenic strains (*Aeromonas veronii*, *Vibrio alginolyticus*, *Vibrio harveyi*, and *Vibrio anguillarum*) in aquaculture fish using the double layer method in TSA culture medium, and biochemical tests (Gram stain, oxidase test, catalase test) for phenotypic characterization of the bacterial strains isolated from the reared *S. dumerili* larvae. (+) inhibition zone up to 10 mm, (++) 11–20 mm, (+++) 21–30 mm, (++++) greater than 30 mm, and (-) absence of inhibition zone.

Pathogenic Strain							
Bacterial Strain	*Aeromonas veronii*	*Vibrio alginolyticus*	*Vibrio harveyi*	*Vibrio anguillarum*	Gram	Oxidase	Catalase
*Vibrio parahaemolyticus* strain VP1	-	++	++++	++++	-	+	+
*Vibrio alginolyticus* strain ZJ-T	++++	++++	++++	++++	-	+	+
*Vibrio alginolyticus* strain Xmb025	+++	+++	++++	+++	-	+	+
*Vibrio alginolyticus* strain Xmb019	+++	++++	+++	+++	-	+	+
Uncultured *Vibrio* sp. clone HH101352	++++	+++	++++	++++	-	+	+
Uncultured *Vibrio* sp. clone HH101334	++++	+++	-	+++	-	+	+
Uncultured *Vibrio* sp. clone C0A05-1	+++	+++	+++	+++	-	+	+
Uncultured *Vibrio* sp. clone HH101375	+++	++++	-	-	-	+	+
*Vibrio* sp. PSMJVIT3	++++	+++	+++	+++	-	+	+
*Vibrio* sp. strain JLT194	++++	++++	+++	++++	-	+	+
*Vibrio* sp. CN87	+++	++++	++++	++++	-	+	+

**Table 5 microorganisms-11-01889-t005:** Antimicrobial activity against pathogenic strains in aquaculture fish using the double layer method, in Marine Agar culture medium, of the bacterial strains isolated from reared *S. dumerili* larvae and *Artemia* sp. (+) inhibition zone up to 5 mm; (++) 5–10 mm and (-) absence of inhibition zone (^1^ isolated strains from homogenated larvae, ^2^ isolated strains from homogenated *Artemia* sp.).

	Pathogenic Strain			
Bacterial Strain	*Aeromonas veronii*	*Vibrio harveyi*	*Vibrio anguillarum*	*Vibrio alginolyticus*
*Phaeobacter gallaeciensis* JL2886 ^1^	+	+	-	-
*Phaeobacter* sp. ^1^	+	+	++	+
*Ruegeria* sp.InS-296 ^1^	++	++	++	++
*Rhodobacter* sp. 1–5 ^2^	+	+	-	-

## Data Availability

The data presented in this study are available upon request from the corresponding author. The data are not publicly available due to short time margins.

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
