# Peer review of "Characterization of Host-Associated Microbiota and Isolation of Antagonistic Bacteria from Greater Amberjack (Seriola dumerili, Risso, 1810) Larvae"

_microorganisms, 2023, doi:10.3390/microorganisms11081889_

Round 1

Reviewer 1 Report (Previous Reviewer 2)

I have recently completed a review of the manuscript titled "Characterization of host-associated microbiota and isolation of antagonistic bacteria from greater amberjack (Seriola dumerili, Risso, 1810) larvae." (microorganisms-2369315). The authors have adequately addressed all of my previous comments. Although they have not made changes to the manuscript, I am satisfied with their responses and accept them.  

Minor editing of English language required

Author Response

Reviewer 1. Comment. Minor editing of English language requires.

ASNWER. Thank you for your comments. We found some few errors:

Line 39. there is still a lack of information… The word a was inserted.

Line 52. through host-selective processes … a hyphen was inserted.

Line 76. The use of copepods… The word The was inserted.

Line 77. resulted in an improvement in bone development and a decrease in the appearance of… instead …of resulted in an improvement of bone development and decrease in the appearance of..

Line 146. using the DESeq2 tool. The word the was inserted.

Line 191. Dr J. Bagge. The period after Dr was deleted.

Line 195. Dr Pantelis Katharios. The period after Dr was deleted.

Line 202. For each test, the inhibition…. A comma was inserted after the word test.

Line 213. The lowest ASV richness. The final s in ASV was deleted.

Lines 298-300. The rest were nine Rhodobacteraceae, three Pseudoalteromonadaceae, three Alteromonadaceae... instead of: The rest were 9 Rhodobacteraceae, 3 Pseudoalteromonadaceae, 3 Alteromonadaceae…

Lines 308. The isolate from Halomonadaceae family matched to Halomonas sp. strain instead of: The isolate from Halomonadaceae family was matching to Halomonas sp. strain.

Lines 418-419. Firmicutes relative abundance differentiated at 31 dah… instead of: the relative abundance of Firmicutes differentiated at 31 dah.

Reviewer 2 Report (New Reviewer)

In the article 'Characterization of host-associated microbiota and isolation of antagonistic bacteria from greater amberjack (Seriola dumerili, Risso, 1810) larvae,' Vasiliki Paralika et al. have conducted a comprehensive investigation into the microbiota present in Seriola dumerili larvae. This manuscript delves into a captivating topic that is sure to engage readers of Microorganisms.

The introduction is well-documented, and the methods are meticulously described, encompassing a series of experiments aimed at unraveling the role of the microbiota in larval development. The experiments have been thoughtfully designed, and the references provided are both accurate and up-to-date.

In light of these findings, I have several inquiries regarding the authors' results. They report a significantly high abundance of genes associated with the perosamine synthetase (K13010) pathway in the copepods group at both time points, 9 and 31. However, this prevalence does not correspond to the relative abundance or diversity indexes. I am curious about the potential factors contributing to this discrepancy.

Furthermore, the author employs the V4 amplicon for microbiota characterization. In my opinion, a longer amplicon, such as the V3-V4 amplicon, could provide more robust results in microbiota characterization. I am interested in the rationale behind using only the V4 amplicon.

Overall, this work is well-organized, easily comprehensible, and scientifically sound. I highly recommend its publication.

Author Response

Reviewer 2.

Comment 1. In light of these findings, I have several inquiries regarding the authors' results. They report a significantly high abundance of genes associated with the perosamine synthetase (K13010) pathway in the copepods group at both time points, 9 and 31. However, this prevalence does not correspond to the relative abundance or diversity indexes. I am curious about the potential factors contributing to this discrepancy.

Answer. Thank you for your comments. We are not very sure what the reviewer means that the prevalence does not correspond to the relative abundance or diversity indexes. The gene abundance analysis is coming from the predictive functionality analysis using the PICRUSt2 software. The software predicts based on the taxa abundance the potential functionality of the taxa present in the gut based on their microbial composition. However, such predictive analysis does not provide information from which bacterial taxa this specific function is coming from. Therefore, we cannot be sure in which certain taxa present in the copepod treatment are responsible for the prevalence of this gene.

Comment 2. Furthermore, the author employs the V4 amplicon for microbiota characterization. In my opinion, a longer amplicon, such as the V3-V4 amplicon, could provide more robust results in microbiota characterization. I am interested in the rationale behind using only the V4 amplicon.

Answer. It is true that V3-V4 is a widely used amplicon, used in many studies. However, V4 is also a very well and widely accepted amplicon for characterization of the fish microbiota. We have added two references [33-34] in the manuscript in the methods section (line 123) to verify that V4 has been used in similar studies recently (2020 and 2022).

This manuscript is a resubmission of an earlier submission. The following is a list of the peer review reports and author responses from that submission.

Round 1

Reviewer 1 Report

The authors of the work “Characterization of host-associated microbiota and isolation of antagonistic bacteria from greater amberjack (Seriola dumerili, Risso, 1810) larvae” carry out the characterization of the greater amberjack microbiome and the isolation of “antagonistic” species. The work is well detailed and the discussion emanates from the results that the authors present. However, I have a few comments to make:

1. It would be recommendable that the authors define “antagonistic” bacteria. Although its meaning can be inferred by reading the work, it should make it easier to understand.

2. The work lacks, significantly, in terms of the number of replicates in the analysis of the microbiome (even when a pool of three larvae is made, all of them are obtained from the same tank at the same time, so they are the same biological replicate -as in bacterial cultures there are billions of bacteria in 50-100 mL and are considered as a single replicate). Statistical significance is lost by using two replicates for each sampling point. They should have taken at least 3.

3. Figure 1 looks blurry. Please include a higher quality version.

4. Why were the cultures kept with the copepods until day 42 and only until day 31 with the controls? What are you trying to show by indicating that they stayed longer?

5. Although the used copepods are a natural component of the Seriola dumerili diet, as the authors indicate, it is difficult to culture. Wouldn't it have been more interesting to use another, more economical type of food that would reduce production costs?

6. Since the isolation of "antagonistic" species aims to identify possible probiotics, a survival analysis of Seriola durmerili larvae with, at least, some of the isolated species is missing. Likewise, the influence of these probiotics on the fish microbiome could also be characterized. Although these works can be reserved for future researches, they would make this work more compact and coherent. Instead, they have simply been presented with two characterizations with little connection between them. Likewise, no data is shown that validates the characterization of the microbiome, remaining only the computer analysis.

Reviewer 2 Report

The present investigation aimed to explore the diversity of bacterial communities in amberjack larvae that were either fed copepods or not. The experiment was carried out with four experimental units, two of which served as the control group and were given rotifers (Brachionus sp.) from 3-27 dah, newly hatched Artemia sp. nauplii from 10-23 dah, enriched Artemia sp. metanauplii from 17 dah until the end of the experiment, and formulated feed from 23 dah onwards. The other two units, the copepod group, were also fed newly hatched copepod nauplii (Acartia tonsa) from 3-17 dah. Then, an in vitro antagonistic test was conducted to examine the bacteria isolated from greater amberjack larvae and live feed against four pathogenic bacteria.

The first part of this study was conducted to investigate the variation of bacterial community composition associated with Amberjack larvae fed or not with copepods. The primary critique pertains to the experimental methodology. Utilizing only two replicates (n=2) comprised of combined samples from three larvae each is inadequate for demonstrating significant differences between various treatments. Consequently, Figure 1 (A and B) exhibits considerable standard error, rendering it difficult to discern any significant variation.

Another significant concern pertains to the choice of feeding protocol involving copepods. It is widely acknowledged that utilizing copepods as a food source and their regular cultivation can be costly and unpredictable. Therefore, exploring alternative feeding strategies is imperative.

The gut microbiota in fish can be influenced by their diet, and varying feeding strategies used to rear greater amberjack larvae and juveniles could potentially impact their susceptibility during early developmental stages. It is currently unknown how microbial communities evolve during the development of greater amberjack under different feeding protocols, and examining this could yield valuable information for the development of improved rearing techniques. However, the findings presented in the study indicate that the core microbiota of both groups (fed or not with copepods) was comparable, suggesting that the experimental design involving feeding with or without copepods requires further clarification.

An experiment should be conducted to evaluate and compare growth performance and immune potentiating activities of feeding protocol on greater amberjack larvae.

L205-206: “The microbial richness, as measured by linear mixed effect model analysis, significantly increased over time (p < 0.05; Table 1)”. However, table 1 showed that diet had a significant effect on both richness and (Shannon Index H’), in contrast to age which had no significant effects.

During the experiment, a gradual increase was observed in richness not for diversity; however, it has been mentioned that a gradual increase was observed in both richness and diversity L212-213.

The second part of this study was conducted for “in vetro inhibition test” of bacteria isolated from greater amberjack larvae and live feed. However, antagonistic inhibition test was conducted against four pathogenic bacteria of Vibrio anguillarum type strain LMG 4437, isolated from Atlantic cod; Vibrio alginolyticus type strain V2 isolated from

Dentex dentex; Vibrio harveyi type strain VH2, isolated from farmed juvenile Seriola dumerili and Aeromonas veronii biovar sobria isolated from farmed European seabass in the Mediterranean Sea.

An experiment should be conducted to test the pathogenicity and mortalities effects of these pathogens on greater amberjack larvae.

Minor editing of English language required